# Continual Learning with Adaptive Weights (CLAW)

**Tameem Adel**
Department of Engineering, University of Cambridge
`tah47@cam.ac.uk`

**Han Zhao**
Carnegie Mellon University
`han.zhao@cs.cmu.edu`

**Richard E. Turner**
Department of Engineering, University of Cambridge
Microsoft Research
`ret26@cam.ac.uk`

## Abstract

Approaches to continual learning aim to successfully learn a set of related tasks that arrive in an online manner. Recently, several frameworks have been developed which enable deep learning to be deployed in this learning scenario. A key modelling decision is to what extent the architecture should be shared across tasks. On the one hand, separately modelling each task avoids catastrophic forgetting but it does not support transfer learning and leads to large models. On the other hand, rigidly specifying a shared component and a task-specific part enables task transfer and limits the model size, but it is vulnerable to catastrophic forgetting and restricts the form of task-transfer that can occur. Ideally, the network should adaptively identify which parts of the network to share in a data driven way. Here we introduce such an approach called Continual Learning with Adaptive Weights (CLAW), which is based on probabilistic modelling and variational inference. Experiments show that CLAW achieves state-of-the-art performance on six benchmarks in terms of overall continual learning performance, as measured by classification accuracy, and in terms of addressing catastrophic forgetting.

## 1 Introduction

Continual learning (CL), sometimes called lifelong or incremental learning, refers to an online framework where the knowledge acquired from learning tasks in the past is kept and accumulated so that it can be reused in the present and future. Data belonging to different tasks could potentially be non i.i.d. (Schlimmer & Fisher, 1986; Sutton & Whitehead, 1993; Ring, 1997; Schmidhuber, 2013; Nguyen et al., 2018; Schmidhuber, 2018). A continual learner must be able to learn a new task, crucially, without forgetting previous tasks (Ring, 1995; Srivastava et al., 2013; Schwarz et al., 2018; Serra et al., 2018; Hu et al., 2019). In addition, CL frameworks should continually adapt to any domain shift occurring across tasks. The learning updates must be incremental – i.e, the model is updated at each task only using the new data and the old model, without access to all previous data (from earlier tasks) – due to speed, security and privacy constraints. A compromise must be found between adapting to new tasks and enforcing stability to preserve knowledge from previous tasks. Excessive adaptation could lead to inadvertent forgetting of how to perform earlier tasks. Indeed, catastrophic forgetting is one of the main pathologies in continual learning (McCloskey & Cohen, 1989; Ratcliff, 1990; Robins, 1993; 1995; French, 1999; Pape et al., 2011; Goodfellow et al., 2014a; Achille et al., 2018; Kemker et al., 2018; Kemker & Kanan, 2018; Diaz-Rodriguez et al., 2018; Zeno et al., 2018; Ahn et al., 2019; Parisi et al., 2019; Pfulb & Gepperth, 2019; Rajasegaran et al., 2019).

Many approaches to continual learning employ an architecture which is divided *a priori* into (i) a slowly evolving, global part; and (ii) a quickly evolving, task-specific, local part. This is one way to enable multi-task transfer whilst mitigating catastrophic forgetting, which has proven to be effective (Rusu et al., 2016b; Fernando et al., 2017; Yoon et al., 2018), albeit with limitations. Specifying *a priori* the shared global, and task-specific local parts in the architecture restricts flexibility. As

more complex and heterogeneous tasks are considered, one would like a more flexible, data-driven approach to determine the appropriate amount of sharing across tasks. Here, we aim at automating the architecture adaptation process so that each neuron of the network can either be kept intact, i.e. acting as global, or adapted to the new task locally. Our proposed variational inference framework is flexible enough to learn the range within which the adaptation parameters can vary. We introduce for each neuron one binary parameter controlling whether or not to adapt, and two parameters to control the magnitude of adaptation. All parameters are learnt via variational inference. We introduce our framework as an expansion of the variational continual learning algorithm (Nguyen et al., 2018), whose variational and sequential Bayesian nature makes it convenient for our modelling and architecture adaptation procedure. Our modelling ideas can also be applied to other continual learning frameworks, see the Appendix for a brief discussion.

We highlight the following contributions: (1) A modelling framework which flexibly automates the adaptation of local and global parts of the (multi-task) continual architecture. This optimizes the tradeoff between mitigating catastrophic forgetting and improving task transfer. (2) A probabilistic variational inference algorithm which supports incremental updates with adaptively learned parameters. (3) The ability to combine our modelling and inference approaches without any significant augmentation of the architecture (no new neurons are needed). (4) State-of-the-art results in six experiments on five datasets, which demonstrate the effectiveness of our framework in terms of overall accuracy and reducing catastrophic forgetting.

## 2 Background on Variational Continual Learning (VCL)

In this paper, we use Variational Continual Learning (VCL, Nguyen et al., 2018) as the underlying continual learning framework. However, our methods apply to other frameworks, see Appendix (Section A.1). VCL is a variational Bayesian framework where the posterior of the model parameters $\boldsymbol{\theta}$ is learnt and updated continually from a sequence of $T$ datasets, $\{\boldsymbol{x}_t^{(n)}, \boldsymbol{y}_t^{(n)}\}_{n=1}^{N_t}$, where $t = 1, 2, \ldots, T$ and $N_t$ is the size of the dataset associated with the $t$-th task. More specifically, denote by $p(\boldsymbol{y}|\boldsymbol{\theta}, \boldsymbol{x})$ the probability distribution returned by a discriminative classifier with input $\boldsymbol{x}$, output $\boldsymbol{y}$ and parameters $\boldsymbol{\theta}$. For $\mathcal{D}_t = \{\boldsymbol{y}_t^{(n)}\}_{n=1}^{N_t}$, we approximate the intractable posterior $p(\boldsymbol{\theta}|\mathcal{D}_{1:t})$ after observing the first $t$ datasets via a tractable variational distribution $q_t$ as:[1]

$$\mathbf{q}_t(\boldsymbol{\theta}) \approx \frac{1}{Z_t}\mathbf{q}_{t-1}(\boldsymbol{\theta})\, p(\mathcal{D}_t|\boldsymbol{\theta}), \tag{1}$$

where $\mathbf{q}_0$ is the prior $p$, $p(\mathcal{D}_t|\boldsymbol{\theta}) = \prod_{n=1}^{N_t} p(\boldsymbol{y}_t^{(n)}|\boldsymbol{\theta}, \boldsymbol{x}_t^{(n)})$, and $Z_t$ is the normalizing constant which does not depend on $\boldsymbol{\theta}$ but only on the data $\mathcal{D}$. This framework allows the approximate posterior $\mathbf{q}_t(\boldsymbol{\theta})$ to be updated incrementally from the previous approximate posterior $\mathbf{q}_{t-1}(\boldsymbol{\theta})$ in an online fashion. In VCL, the approximation in (1) is performed by minimizing the following KL-divergence over a family $\mathcal{Q}$ of tractable distributions:

$$\mathbf{q}_t(\boldsymbol{\theta}) = \underset{\mathbf{q}\in\mathcal{Q}}{\operatorname{argmin}} \operatorname{KL}\Big(\mathbf{q}(\boldsymbol{\theta}) \,\|\, \frac{1}{Z_t}\mathbf{q}_{t-1}(\boldsymbol{\theta})\, p(\mathcal{D}_t|\boldsymbol{\theta})\Big). \tag{2}$$

This framework can be enhanced to further mitigate catastrophic forgetting by using a coreset (Nguyen et al., 2018), i.e. a representative set of data from previously observed tasks that can serve as memory and can be revisited before making a decision. As discussed in the Related Work, this leads to overhead costs of memory and optimisation (selecting most representative data points). Previous work on VCL considered simple models without automatic architecture building or adaptation.

## 3 Our CLAW Approach

In earlier CL approaches, the parts of the network architecture that are shared among the learnt tasks are designated *a priori*. To alleviate this rigidity and to effectively balance adaptation and stability, we propose a multi-task, continual model in which the adaptation of the architecture is data-driven by learning which neurons need to be adapted as well as the maximum adaptation capacity for each. All the model parameters (including those used for adaptation) are estimated via an efficient variational

---

[1]Here we suppress the dependence on the inputs in $p(\boldsymbol{\theta}|\mathcal{D}_{1:t})$ and $p(\mathcal{D}_t|\boldsymbol{\theta})$ to lighten notation.

inference algorithm which incrementally learns from data of the successive tasks, without a need to store (nor generate) data from previous tasks and with no expansion in the network size.

## 3.1 MODELLING

With model parameters $\boldsymbol{\theta}$, the overall variational objective we aim at maximising at task with index $\mathbf{t}$ is equivalent to the following online marginal likelihood:

$$\mathcal{L}(\boldsymbol{\theta}) = -\mathrm{KL}\Big(\mathbf{q}_t(\boldsymbol{\theta}) \parallel \mathbf{q}_{t-1}(\boldsymbol{\theta})\Big) + \sum_{n=1}^{N_t} \mathbb{E}_{\mathbf{q}_t(\boldsymbol{\theta})}\big[\log p(\boldsymbol{y}^{(n)}|\boldsymbol{x}^{(n)}, \boldsymbol{\theta})\big]. \tag{3}$$

We propose a framework where the architecture, whose parameters are $\boldsymbol{\theta}$, is flexibly adapted based on the available tasks, via a learning procedure that will be described below. With each task, we automate the adaptation of the neuron contributions. Both the adaptation decisions (i.e. whether or not to adapt) and the maximum allowed degree of adaptation for every neuron are learnt. We refer to the binary adaptation variable as $\boldsymbol{\alpha}$. There is another variable $\mathbf{s}$ that is learnt in a multi-task fashion to control the maximum degree of adaptation, such that the expression $\mathbf{b} = \frac{\mathbf{s}}{1+\mathrm{e}^{-\mathbf{a}}} - 1$ limits how far the task-specific weights can differ from the global weights, in case the respective neuron is to be adapted. The parameter $\mathbf{a}$ depicts unconstrained adaptation, as described later.[2]

We illustrate the proposed model to perform this adaptation by learning the probabilistic contributions of the different neurons within the network architecture on a task-by-task basis. We follow this with the inference details. Steps of the proposed modeling are listed as follows:

- For a task $T$, the classifier that we are modeling outputs: $\sum_{n=1}^{N_T}\big[\log p(\boldsymbol{y}^{(n)}|\boldsymbol{x}^{(n)}, \mathbf{w}^T)\big]$ .
- The task-specific weights $\mathbf{w}^T$ can be expressed in terms of their global counterparts as follows:

$$\mathbf{w}^T = (1 + \mathbf{b}^T\boldsymbol{\alpha}^T) \circ \mathbf{w}. \tag{4}$$

  The symbol $\circ$ denotes an element-wise (Hadamard) multiplication.
- For each task $T$ and each neuron $\mathbf{j}$ at layer $\mathbf{i}$, $\boldsymbol{\alpha}_{\mathbf{i,j}}^T$ is a binary variable which indicates whether the corresponding weight is adapted ($\boldsymbol{\alpha}_{\mathbf{i,j}}^T = 1$) or unadapted ($\boldsymbol{\alpha}_{\mathbf{i,j}}^T = 0$). Initially assume that the adaptation probability $\boldsymbol{\alpha}_{\mathbf{i,j}}^T$ follows a Bernoulli distribution with probability $\mathbf{p_{i,j}}$[3], $\boldsymbol{\alpha}_{\mathbf{i,j}}^T \sim$ Bernoulli($\mathbf{p_{i,j}}$). Since this Bernoulli is not straightforward to optimise, and to adopt a scalable inference procedure based on continuous latent variables, we replace this Bernoulli with a Gaussian that has an equivalent mean and variance from which we draw $\boldsymbol{\alpha}_{\mathbf{i,j}}^T$. For the sake of attaining higher fidelity than what is granted by a standard Gaussian, we base our inference on a variational Gaussian estimation. Though in a context different from continual learning and with different estimators, the idea of replacing Bernoulli with an equivalent Gaussian has proven to be effective with dropout (Srivastava et al., 2014; Kingma et al., 2015).

  The approximation of the Bernoulli distribution by the corresponding Gaussian distribution is achieved by matching the mean and variance. The mean and variance of the Bernoulli distribution are $\mathbf{p_{i,j}}$, $\mathbf{p_{i,j}}(1 - \mathbf{p_{i,j}})$, respectively. A Gaussian distribution with the same mean and variance is used to fit $\boldsymbol{\alpha}_{\mathbf{i,j}}^T$.

$$\boldsymbol{\alpha}_{\mathbf{i,j}}^T \sim \mathcal{N}(\mathbf{p_{i,j}}, \mathbf{p_{i,j}}(1 - \mathbf{p_{i,j}})). \tag{5}$$

- The variable $\mathbf{b}^T$ controls the strength of the adaptation and it limits the range of adaptation via:

$$1 + \mathbf{b}^T = \frac{\mathbf{s}}{1 + \mathrm{e}^{-\mathbf{a}^T}}. \tag{6}$$

  So that the maximum adaptation is $\mathbf{s}$. The variable $\mathbf{a}^T$ is an unconstrained adaptation value, similar to that in (Swietojanski & Renals, 2014). The addition of 1 is to facilitate the usage of a probability distribution while still keeping an adaptation range allowing for the attenuation or amplification of each neuron's contribution.

---

[2]We learn parameters $\boldsymbol{\alpha}$, $\mathbf{s}$ and $\mathbf{a}$, with corresponding $\mathbf{b}$, for each neuron but we avoid using the neuron indices here, as well as in other locations like (4), (6) and (7), for better readability.

[3]There can be a parameter $\mathbf{p_i}$ per layer $\mathbf{i}$ instead of one $\mathbf{p_{i,j}}$ for each neuron $\mathbf{j}$ at each layer $\mathbf{i}$, but we opt for the latter for the sake of gaining further adaptation flexibility.

- Before facing the first dataset and learning task $\mathbf{t} = 1$, the prior on the weights $\mathbf{q}_0(\mathbf{w}) = \mathbf{p}(\mathbf{w})$ is chosen to be a log-scale prior, which can be expressed as: $\mathbf{p}(\log|\mathbf{w}|) \propto \mathbf{c}$, where $\mathbf{c}$ is a constant. The log-scale prior can alternatively be described as:

$$\mathbf{p}(|\mathbf{w}|) \propto \frac{1}{|\mathbf{w}|}. \tag{7}$$

At a high level, adapting neuron contributions can be seen as a generalisation of attention mechanisms in the context of continual learning. Applying this adaptation procedure to the input leads to an attention mechanism. However, our approach is more general since we do not apply it only to the very bottom (i.e. input) layer, but throughout the whole network. We next show how our variational inference mechanism enables us to learn the adaptation parameters.

## 3.2 Inference

We describe the details related to the proposed variational inference mechanism. The adaptation parameters are included within the variational parameters.

The (unadapted version of the) model parameters $\boldsymbol{\theta}$ consist of the weight vectors $\mathbf{w}$. To automate adaptation, we perform inference on $\mathbf{p_{i,j}}$, which would have otherwise been a hyperparameter of the prior (Louizos et al., 2017; Molchanov et al., 2017; Ghosh et al., 2018). Multiplying $\mathbf{w}$ by $(1 + \mathbf{b}\boldsymbol{\alpha})$ where $\boldsymbol{\alpha}$ is distributed according to (5), then from (4) with random noise variable $\boldsymbol{\epsilon} \sim \mathcal{N}(0,1)$:

$$\mathbf{w_{i,j}^T} = \boldsymbol{\gamma_{i,j}} \left( 1 + \mathbf{b_{i,j}}\mathbf{p_{i,j}} + \mathbf{b_{i,j}}\sqrt{\mathbf{p_{i,j}}(1 - \mathbf{p_{i,j}})}\boldsymbol{\epsilon} \right),$$

$$\mathbf{q}(\mathbf{w_{i,j}} \mid \boldsymbol{\gamma_{i,j}}) \sim \mathcal{N}\left( \boldsymbol{\gamma_{i,j}}(1 + \mathbf{b_{i,j}}\mathbf{p_{i,j}}), \mathbf{b_{i,j}^2}\boldsymbol{\gamma_{i,j}^2}\mathbf{p_{i,j}}(1 - \mathbf{p_{i,j}}) \right). \tag{8}$$

From (7) and (8), the corresponding KL-divergence between the variational posterior of $\mathbf{w}$, $\mathbf{q}(\mathbf{w}|\boldsymbol{\gamma})$ and the prior $\mathbf{p}(\mathbf{w})$ is as follows. The subscripts are removed when $\mathbf{q}$ in turn is used as a subscript for improved readability. The variational parameters are $\boldsymbol{\gamma_{i,j}}$ and $\mathbf{p_{i,j}}$.

$$\mathrm{KL}\left( \mathbf{q}(\mathbf{w_{i,j}}|\boldsymbol{\gamma_{i,j}}) \parallel \mathbf{p}(\mathbf{w_{i,j}}) \right) = \mathbb{E}_{\mathbf{q}(\mathbf{w}|\boldsymbol{\gamma})} \log[\mathbf{q}(\mathbf{w_{i,j}}|\boldsymbol{\gamma_{i,j}})/\mathbf{p}(\mathbf{w_{i,j}})] =$$

$$\mathbb{E}_{\mathbf{q}(\mathbf{w}|\boldsymbol{\gamma})} \log \mathbf{q}(\mathbf{w_{i,j}}|\boldsymbol{\gamma_{i,j}}) - \mathbb{E}_{\mathbf{q}(\mathbf{w}|\boldsymbol{\gamma})} \log \mathbf{p}(\mathbf{w_{i,j}}) = -\mathbf{H}(\mathbf{q}(\mathbf{w_{i,j}}|\boldsymbol{\gamma_{i,j}})) - \mathbb{E}_{\mathbf{q}(\mathbf{w}|\boldsymbol{\gamma})} \log \mathbf{p}(\mathbf{w_{i,j}}) \tag{9}$$

$$= -0.5\left( 1 + \log(2\pi) + \log(\mathbf{b_{i,j}^2}\mathbf{p_{i,j}}(1 - \mathbf{p_{i,j}})) \right) - \mathbb{E}_{\mathbf{q}(\mathbf{w}|\boldsymbol{\gamma})} \log \frac{1}{|\boldsymbol{\epsilon}|} \tag{10}$$

$$= -\log \mathbf{b_{i,j}} - 0.5 \log \mathbf{p_{i,j}} - 0.5 \log(1 - \mathbf{p_{i,j}}) + \mathbf{c} + \mathbb{E}_{\mathbf{q}(\mathbf{w}|\boldsymbol{\gamma})} \log |\boldsymbol{\epsilon}|, \tag{11}$$

where the switch from (9) to (10) is due to the entropy computation (Bernardo & Smith, 2000) of the Gaussian $\mathbf{q}(\mathbf{w_{i,j}}|\boldsymbol{\gamma_{i,j}})$ defined in (8). The switch from (10) to (11) is due to using a log-scale prior, similar to Appendix C in (Kingma et al., 2015) and to Section 4.2 in (Molchanov et al., 2017). $\mathbb{E}_{\mathbf{q}(\mathbf{w}|\boldsymbol{\gamma})} \log |\boldsymbol{\epsilon}|$ is computed via an accurate approximation similar to equation (14) in (Molchanov et al., 2017), with slightly different values of $k_1$, $k_2$ and $k_3$. This is a very close approximation via numerically pre-computing $\mathbb{E}_{\mathbf{q}(\mathbf{w}|\boldsymbol{\gamma})} \log |\boldsymbol{\epsilon}|$ using a third degree polynomial (Kingma et al., 2015; Molchanov et al., 2017).

This is the form of the KL-divergence between the approximate posterior after the first task and the prior. Afterwards, it is straightforward to see how this KL-divergence applies for the subsequent tasks in a manner similar to (2), but while taking into account the new posterior form and original prior.

The KL-divergence expression derived in (11) is to be minimised. By minimising (11) with respect to $\mathbf{p_{i,j}}$ and then using samples from the respective distributions to assign values to $\boldsymbol{\alpha_{i,j}}$, adapted contributions of each neuron $\mathbf{j}$ at each layer $\mathbf{i}$ of the network are learnt per task. Values of $\mathbf{p_{i,j}}$ are constrained between 0 and 1 during training via projected gradient descent.

---

**Algorithm 1** Continual Learning with Adaptive Weights (`CLAW`)

---

**Input:** A sequence of $T$ datasets, $\{\boldsymbol{x}_t^{(n)}, \boldsymbol{y}_t^{(n)}\}_{n=1}^{N_t}$, where $t = 1, 2, \ldots, T$ and $N_t$ is the size of the dataset associated with the $t$-th task.
**Output:** $\mathbf{q}_t(\boldsymbol{\theta})$, where $\boldsymbol{\theta}$ are the model parameters.
   Initialise all $\mathbf{p}(|\mathbf{w_{i,j}}|)$ with a log-scale prior, as in (7).
   **for** $t = 1 \ldots T$ **do**
      Disclose the dataset $\{\boldsymbol{x}_t^{(n)}, \boldsymbol{y}_t^{(n)}\}_{n=1}^{N_t}$ for the current task **t**.
      **for i** $= 1 \ldots \#$ layers **do**
         **for j** $= 1 \ldots \#$ neurons at layer **i do**
            Compute $\mathbf{p_{i,j}}$ using stochastic gradient descent on (11).
            Compute $\mathbf{s_{i,j,t}}$ using (13).
            Update the corresponding general value $\mathbf{s_{i,j}}$ using (14).
         **end for**
      **end for**
   **end for**

---

### 3.2.1 LEARNING THE MAXIMUM ADAPTATION VALUES

Using (6) to express the value of $\mathbf{b_{i,j}}$, and neglecting the constant term therein since it does not affect the optimisation, the KL-divergence in (11) is equivalent to:

$$\mathrm{KL}\Big(\mathbf{q}(\mathbf{w_{i,j}}|\boldsymbol{\gamma_{i,j}}) \,\|\, \mathbf{p}(\mathbf{w_{i,j}})\Big) \approx$$
$$-\log \mathbf{s_{i,j}} + \log(1 + e^{-\mathbf{a_{i,j}}}) - 0.5 \log \mathbf{p_{i,j}} - 0.5 \log(1 - \mathbf{p_{i,j}}) + \mathbf{c} + \mathbb{E}_{\mathbf{q}(\mathbf{w}|\boldsymbol{\gamma})} \log |\boldsymbol{\epsilon}|. \quad (12)$$

Values of $\mathbf{a_{i,j}}$ are learnt by minimising (12) with respect to $\mathbf{a_{i,j}}$. This subsection explains how to learn the maximum adaptation variable $\mathbf{s_{i,j}}$. Values of the maximum $\mathbf{s_{i,j}}$ of the logistic function defined in (6) are learnt from multiple tasks. For each neuron **j** at layer **i**, there is a general value $\mathbf{s_{i,j}}$ and another value that is specific for each task **t**, referred to as $\mathbf{s_{i,j,t}}$. This is similar to the meta-learning procedure proposed in (Finn et al., 2017). The following procedure to learn **s** is performed for each task **t** such that: (i) the optimisation performed to learn a task-specific value $\mathbf{s_{i,j,t}}$ benefits from the warm initialisation with the general value $\mathbf{s_{i,j}}$ rather than a random initial condition; and then (ii) the new information obtained from the current task **t** is reflected back to update the general value $\mathbf{s_{i,j}}$.

- First divide the sample $N_\mathbf{t}$ into two halves. For the first half, depart from the general value of $\mathbf{s_{i,j}}$ as an initial condition, and use the assigned data examples from task **t** to learn the task-specific values $\mathbf{s_{i,j,t}}$ for the current task **t**. For neuron **j** at layer **i**, refer to the second term in (3), $\sum_{n=1}^{N_t} \mathbb{E}_{\mathbf{q}_t(\boldsymbol{\theta})}\big[\log p(\boldsymbol{y}^{(n)}|\boldsymbol{x}^{(n)}, \boldsymbol{\theta})\big]$ as $\mathbf{f}_t(\boldsymbol{x}, \boldsymbol{y}, \mathbf{s_{i,j}})$. The set of parameters $\boldsymbol{\theta}$ contains **s** as well as other parameters, but we focus here on **s** in the **f** notation since the following procedure is developed to optimise **s**. Also, refer to the loss of the (classification) function **f** as $\mathbf{Err}(\mathbf{f}) = \mathrm{CE}(\mathbf{f}(\boldsymbol{x}, \theta)\|\boldsymbol{y})$, where CE stands for the cross-entropy:

$$\mathbf{s_{i,j,t}} = \mathbf{s_{i,j}} - \frac{2\boldsymbol{\omega}_1}{N_t} \nabla_{\mathbf{s_{i,j}}} \sum_{d=1}^{N_t/2} \mathbf{Err}(\mathbf{f}_t(\boldsymbol{x_d}, \boldsymbol{y_d}, \mathbf{s_{i,j}})). \quad (13)$$

- Now use the second half of the data from task **t** to update the general learnt value $\mathbf{s_{i,j}}$:

$$\mathbf{s_{i,j}} = \mathbf{s_{i,j}} - \frac{2\boldsymbol{\omega}_2}{N_t} \nabla_{\mathbf{s_{i,j}}} \sum_{d=1+N_t/2}^{N_t} \mathbf{Err}(\mathbf{f}_t(\boldsymbol{x_d}, \boldsymbol{y_d}, \mathbf{s_{i,j,t}})). \quad (14)$$

   Where $\boldsymbol{\omega}_1$ and $\boldsymbol{\omega}_2$ are step-size parameters.

When testing on samples from task **t** after having faced future tasks $\mathbf{t} + 1, \mathbf{t} + 2, \ldots$, the value of $\mathbf{s_{i,j}}$ used is the learnt $\mathbf{s_{i,j,t}}$. There is only one value per neuron, so the overhead resulting from storing such values is negligible. The key steps of the algorithm are listed in Algorithm 1.

At task **t**, the algorithmic complexity of a single joint update of the parameters $\boldsymbol{\theta}$ based on the additive terms in (12) is $O(MELD^2)$, where $L$ is the number of layers in the network, $D$ is the

(largest) number of neurons within a single layer, $E$ is the number of samples taken from the random noise variable $\epsilon$, and $M$ is the minibatch size. Each $\boldsymbol{\alpha}$ is obtained by taking one sample from the corresponding $\mathbf{p}$, so that does not result in an overhead in terms of the complexity.

## 4 EXPERIMENTS

Our experiments mainly aim at evaluating the following: (i) the overall performance of the introduced CLAW, depicted by the average classification accuracy over all the tasks; (ii) the extent to which catastrophic forgetting can be mitigated when deploying CLAW; and (iii) the achieved degree of positive forward transfer. The experiments demonstrate the effectiveness of CLAW in achieving state-of-the-art continual learning results measured by classification accuracy and by the achieved reduction in catastrophic forgetting. We also perform ablations in Section D in the Appendix which exhibit the relevance of each of the proposed adaptation parameters.

We perform six experiments on five datasets. The datasets in use are: MNIST (LeCun et al., 1998), notMNIST (Butalov, 2011), Fashion-MNIST (Xiao et al., 2017), Omniglot (Lake et al., 2011) and CIFAR-100 (Krizhevsky & Hinton, 2009). We compare the results obtained by CLAW to six different state-of-the-art continual learning algorithms: the VCL algorithm (Nguyen et al., 2018) (original form and one with a coreset), the elastic weight consolidation (EWC) algorithm (Kirkpatrick et al., 2017), the progress and compress (P&C) algorithm (Schwarz et al., 2018), the reinforced continual learning (RCL) algorithm (Xu & Zhu, 2018), the one referred to as functional regularisation for continual learning (FRCL) using Gaussian processes (Titsias et al., 2019) and the learn-to-grow (LTG) algorithm (Li et al., 2019b).

### 4.1 OVERALL CLASSIFICATION ACCURACY

Our main metric is the all-important classification accuracy. We consider six continual learning experiments, based on the MNIST, notMNIST, Fashion-MNIST, Omniglot and CIFAR-100 datasets. The introduced CLAW is compared to two VCL versions: VCL with no coreset and VCL with a 200-point coreset assembled by the K-center method (Nguyen et al., 2018), EWC, P&C, RCL, FRCL (its TR version) and LTG[4]. All the reported classification accuracy values reflect the average classification accuracy over all tasks the learner has trained on so far. More specifically, assume that the continual learner has just finished training on a task $t$, then the reported classification accuracy at time $t$ is the average accuracy value obtained from testing on equally sized sets each belonging to one of the tasks $1, 2, \ldots, t$. For all the classification experiments, statistics reported are averages of ten repetitions. Statistical significance and standard error of the average classification accuracy obtained after completing the last two tasks of each experiment are displayed in Section E in the Appendix.

As can be seen in Figure 1, CLAW achieves state-of-the-art classification accuracy in all the six experiments. The minibatch size is 128 for Split MNIST and 256 for all the other experiments. More detailed descriptions of the results of every experiment are given next:

**Permuted MNIST** Using MNIST, Permuted MNIST is a standard continual learning benchmark (Goodfellow et al., 2014a; Kirkpatrick et al., 2017; Zenke et al., 2017). For each task $t$, the corresponding dataset is formed by performing a fixed random permutation process on labeled MNIST images. This random permutation is unique per task, i.e. it differs for each task. For the hyperparameter $\lambda$ of EWC, which controls the overall contribution from previous data, we experimented with two values, $\lambda = 1$ and $\lambda = 100$. We report the latter since it has always outperformed EWC with $\lambda = 1$ in this experiment. EWC with $\lambda = 100$ has also previously produced the best EWC classification results (Nguyen et al., 2018). In this experiment, fully connected single-head networks with two hidden layers are used. There are 100 hidden units in each layer, with ReLU activations. Adam (Kingma & Ba, 2015) is the optimiser used in the 6 experiments with $\eta = 0.001$, $\beta_1 = 0.9$ and $\beta_2 = 0.999$. Further experimental details are given in Section C in the Appendix. Results of the accumulated classification accuracy, averaged over tasks, on a test set are displayed in Figure 1a. After 10 tasks, CLAW achieves significantly (check the Appendix) higher classification results than all the competitors.

---

[4]Whenever there is no validation process performed to indicate the hyperparameter values of competitors or characteristics of neural network architectures, this is done for the sake of comparing on common ground with the best settings, as specified in the respective papers.

**Split MNIST** In this MNIST based experiment, five binary classification tasks are processed in the following sequence: 0/1, 2/3, 4/5, 6/7, and 8/9 (Zenke et al., 2017). The architecture used consists of fully connected multi-head networks with two hidden layers, each consisting of 256 hidden units with ReLU activations. As can be seen in Figure 1b, CLAW achieves the highest classification accuracy.

**Split notMNIST** It contains 400,000 training images, and the classes are 10 characters, from A to J. Each image consists of one character, and there are different font styles. The five binary classification tasks are: A/F, B/G, C/H, D/I, and E/J. The networks used here contain four hidden layers, each containing 150 hidden units with ReLU activations. CLAW achieves a clear improvement in classification accuracy over competitors (Figure 1c).

**Split Fashion-MNIST** Fashion-MNIST is a dataset whose size is the same as MNIST but it is based on different (and more challenging) 10 classes. The five binary classification tasks here are: T-shirt/Trouser, Pullover/Dress, Coat/Sandals, Shirt/Sneaker, and Bag/Ankle boots. The architecture used is the same as in Split notMNIST. In most of the continual learning tasks (including the more significant, later ones) CLAW achieves a clear classification improvement (Figure 1d).

**Omniglot** This is a sequential learning task of handwritten characters of 50 alphabets (a total of over 1,600 characters with 20 examples each) belonging to the Omniglot dataset (Lake et al., 2011). We follow the same way via which this task has been used in continual learning before (Schwarz et al., 2018; Titsias et al., 2019); handwritten characters from each alphabet constitute a separate task. We thus have 50 tasks, which also allows to evaluate the scalability of the frameworks in comparison. The model used is a CNN. To deal with the convolutions in CLAW, we used the idea proposed and referred to as the local reparameterisation trick by Kingma et al. (2014; 2015), where a single global parameter is employed per neuron activation in the variational distribution, rather than employing parameters for every constituent weight element[5]. Further details about the CNN used are given in Section C. The automatically adaptable CLAW achieves better classification accuracy (Figure 1e).

**CIFAR-100** This dataset consists of 60,000 colour images of size $32 \times 32$. It contains 100 classes, with 600 images per class. We use a split version CIFAR-100. Similar to Lopez-Paz & Ranzato (2017), we perform a 20-task experiment with a disjoint subset of five classes per task. CLAW achieves significantly higher classification accuracy (Figure 1f) -also higher than the previous state of the art on CIFAR-100 by Kemker & Kanan (2018). Details of the used CNN are in Section C.

A conclusion that can be taken from Figure 1(a-f) is that CLAW consistently achieves state-of-the-art results (in all the 6 experiments). It can also be seen that CLAW scales well. For instance, the difference between CLAW and the best competitor is more significant with Split notMNIST than it is with the first two experiments, which are based on the smaller and less challenging MNIST. Also, CLAW achieves good results with Omniglot and CIFAR-100.

## 4.2 CATASTROPHIC FORGETTING

To assess catastrophic forgetting, we show how the accuracy on the initial task varies over the course of the training procedure on the remaining tasks (Schwarz et al., 2018). Since Omniglot (and CIFAR-100) contain a larger number of tasks: 50 (20) tasks, i.e. 49 (19) remaining tasks after the initial task, this setting is more relevant for Omniglot and CIFAR-100. We nonetheless display the results for Split MNIST, Split notMNIST, Split Fashion-MNIST, Omniglot and CIFAR-100. As can be seen in Figure 2, CLAW (at times jointly) achieves state-of-the-art performance retention degrees. Among the competitors, P&C and LTG also achieve high performance retention degrees.

An empirical conclusion that can be made out of this and the previous experiment, is that CLAW achieves better overall continual learning results, partially thanks to the way it addresses catastrophic forgetting. The idea of adapting the architecture by adapting the contributions of neurons of each layer also seems to be working well with datasets like Omniglot and CIFAR-100, giving directions for imminent future work where CLAW can be extended for other application areas based on CNNs.

## 4.3 POSITIVE FORWARD TRANSFER

The purpose of this experiment is to assess the impact of learning previous tasks on the current task. In other words, we want to evaluate whether an algorithm avoids negative transfer, by evaluating the

---

[5]For more details, see Section 2.3 in (Kingma et al., 2015).

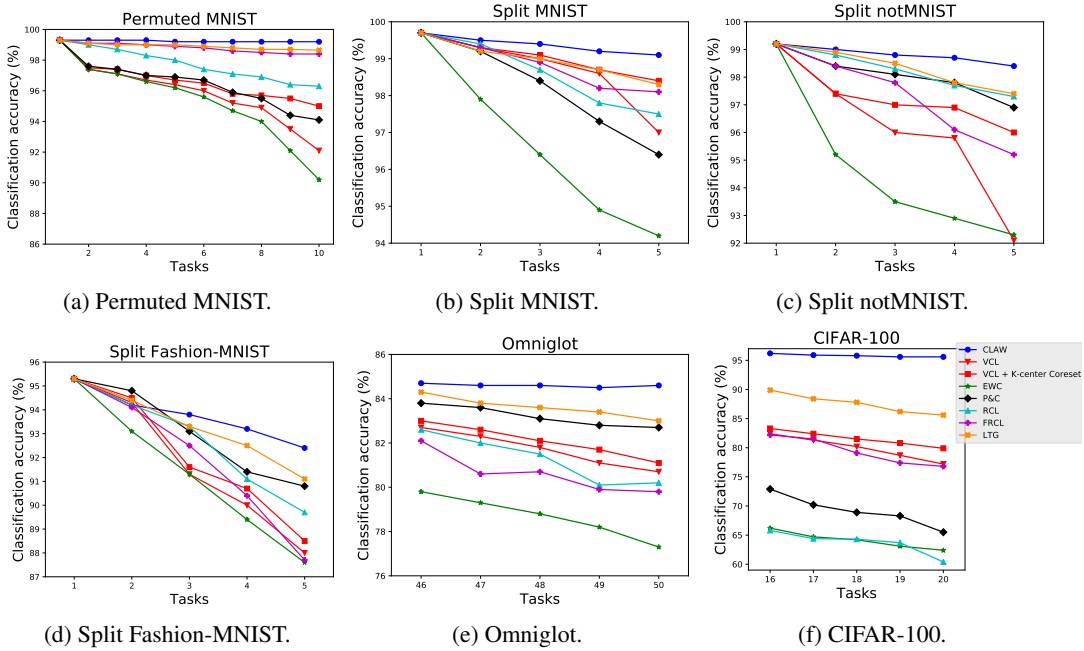

Figure 1: Average test classification accuracy vs. the number of observed tasks in 6 experiments. CLAW achieves significantly higher classification results than the competing continual learning frameworks. Statistical significance values are presented in Section E in the Appendix. The value of $\lambda$ for EWC is 10,000 in (c), and 100 in the other experiments. Best viewed in colour.

relative performance achieved on a unique task after learning a varying number of previous tasks (Schwarz et al., 2018). From Figure 3, we can see that CLAW achieves state-of-the-art results in 4 out of the 5 experiments (at par in the fifth) in terms of avoiding negative transfer.

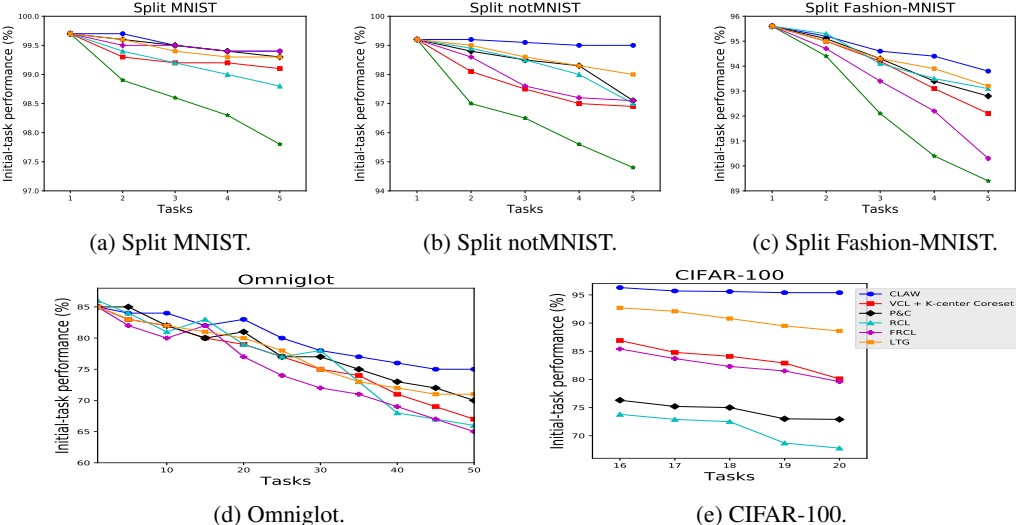

Figure 2: Evaluating catastrophic forgetting by measuring performance retention. Classification accuracy of the initial task is monitored along with the progression of tasks. Results are displayed for five datasets. CLAW is the least forgetful algorithm since performance levels achieved on the initial task do not degrade as much as in the other methods after facing new tasks. The legend and $\lambda$ values for EWC are the same as in Figure 1. Best viewed in colour.

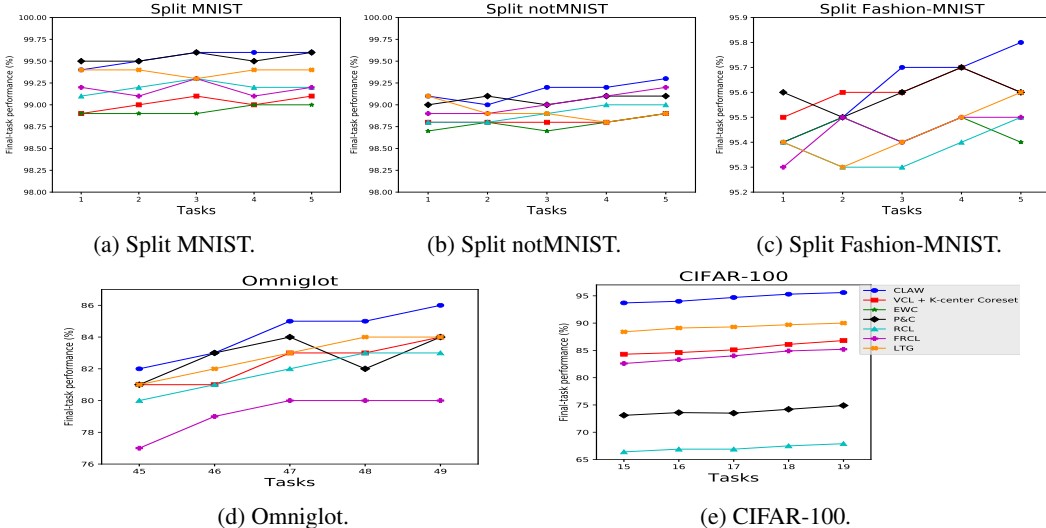

Figure 3: Evaluating Forward transfer, or to what extent a continual learning framework can avoid negative transfer. The impact of learning previous tasks on a specific task (the last task) is inspected and used as a proxy for evaluating forward transfer. This is performed by evaluating the relative performance achieved on a unique task after learning a varying number of previous tasks. This means that the value at x-axis = 1 refers to the learning accuracy of the last task after having learnt solely one task (only itself), the value at 2 refers to the learning accuracy of the last task after having learnt two tasks (an additional previous task), etc. Overall, CLAW achieves state-of-the-art results in 4 out of the 5 experiments (at par in the fifth) in terms of avoiding negative transfer. Best viewed in colour.

## 5    RELATED WORK

We briefly discuss three related approaches to continual learning: (a) regularisation-based, (b) architecture-based and (c) memory-based. We provide more details of related work in Section A in the Appendix. (a) A complementary approach to CLAW is the regularisation-based approach to balance adaptability with catastrophic forgetting: a level of stability is kept via protecting parameters that greatly influence the prediction against radical changes, while allowing the rest of the parameters to change without restriction (Li & Hoiem, 2016; Lee et al., 2017; Zenke et al., 2017; Chaudhry et al., 2018; Kim et al., 2018; Nguyen et al., 2018; Srivastava et al., 2013; Schwarz et al., 2018; Vuorio et al., 2018; Aljundi et al., 2019c). The elastic weight consolidation (EWC) algorithm by Kirkpatrick et al. (2017) is a seminal example, where a quadratic penalty is imposed on the difference between parameter values of the old and new tasks. One limitation is the high level of hand tuning required. (b) The architecture-based approach aims to deal with stability and adaptation issues by a fixed division of the architecture into global and local parts (Rusu et al., 2016b; Fernando et al., 2017; Shin et al., 2017; Kaplanis et al., 2018; Xu & Zhu, 2018; Yoon et al., 2018; Li et al., 2019b). (c) The memory-based approach relies on episodic memory to store data (or pseudo-data) from previous tasks (Ratcliff, 1990; Robins, 1993; 1995; Thrun, 1996; Schmidhuber, 2013; Hattori, 2014; Mocanu et al., 2016; Rebuffi et al., 2017; Kamra et al., 2017; Shin et al., 2017; Rolnick et al., 2018; van de Ven & Tolias, 2018; Wu et al., 2018; Titsias et al., 2019). Limitations include overheads for tasks such as data storage, replay, and optimisation to select (or generate) the points. CLAW can as well be seen as a combination of a regularisation-based approach (the variational inference mechanism) and a modelling approach which automates the architecture building process in a data-driven manner, avoiding the overhead resulting from either storing or generating data points from previous tasks. CLAW is also orthogonal to (and simple to combine with, if needed) memory-based methods.

## 6    CONCLUSION

We introduced a continual learning framework which learns how to adapt its architecture from the tasks and data at hand, based on variational inference. Rather than rigidly dividing the architecture

into shared and task-specific parts, our approach adapts the contributions of each neuron. We achieve that without having to expand the architecture with new layers or new neurons. Results of six different experiments on five datasets demonstrate the strong empirical performance of the introduced framework, in terms of the average overall continual learning accuracy and forward transfer, and also in terms of effectively alleviating catastrophic forgetting.

ACKNOWLEDGMENTS

HZ acknowledges support from the DARPA XAI project, contract#FA87501720152 and an Nvidia GPU grant. RT acknowledges support by Google, Amazon, Improbable and EPSRC grants EP/M0269571 and EP/L000776/1.

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

APPENDIX

We begin by briefly summarising the contents of the Appendix below:

- Related works are described in Section A, followed by a brief discussion on the potential applicability of CLAW to another continual learning (CL) framework in Section A.1.
- In Section E, we provide the statistical significance and standard error of the average classification accuracy results obtained after completing the last two tasks from each experiment.
- Further experimental details are given in Section C.
- In Section D and Figures 4a- 4e, we display the results of performed ablations which manifest the relevance of each adaptation parameter.

## A   MORE RELATED WORK

A complementary approach to CLAW, which could be combined with it, is the regularisation-based approach to balance adaptability with catastrophic forgetting: a level of stability is kept via protecting parameters that greatly influence the prediction against radical changes, while allowing the rest of the parameters to change without restriction (Li & Hoiem, 2016; Vuorio et al., 2018). In (Zenke et al., 2017), the regulariser is based on synapses where an importance measure is locally computed at each synapse during training, based on their respective contributions to the change in the global loss. During a task change, the less important synapses are given the freedom to change whereas catastrophic forgetting is avoided by preventing the important synapses from changing (Zenke et al., 2017). The elastic weight consolidation (EWC) algorithm, introduced by Kirkpatrick et al. (2017), is a seminal example of this approach where a quadratic penalty is imposed on the difference between parameter values of the old and new tasks. One limitation of EWC, which is rather alleviated by using minibatch or stochastic estimates, appears when the output space is not low-dimensional, since the diagonal of the Fisher information matrix over parameters of the old task must be computed, which requires a summation over all possible output labels (Kirkpatrick et al., 2017; Zenke et al., 2017; Schwarz et al., 2018). In addition, the regularisation term involves a sum over all previous tasks with a term from each and a hand-tuned hyperparameter that alters the weight given to it. The accumulation of this leads to a lot of hand-tuning. The work in (Chaudhry et al., 2018) is based on penalising confident fitting to the uncertain knowledge by a maximum entropy regulariser.

Another seminal algorithm based on regularisation, which can be applied to any model, is variational continual learning (VCL) (Nguyen et al., 2018) which formulates CL as a sequential approximate (variational) inference problem. However, VCL has only been applied to simple architectures, not involving any automatic model building or adaptation. The framework in (Lee et al., 2017) incrementally matches the moments of the posterior of a Bayesian neural network that has been trained on the first and then the second task, and so on. Other algorithms pursue regularisation approaches based on sparsity (Srivastava et al., 2013; Kim et al., 2018). For example, the work in (Aljundi et al., 2019c) encourages sparsity on the neuron activations to alleviate catastrophic forgetting. The $l_2$ distance between the top hidden activations of the old and new tasks is used for regularisation in (Jung et al., 2016). This approach has achieved good results, but is computationally expensive due to the necessity of computing at least a forward pass for every new data point through the network representing the old task (Zenke et al., 2017). Other regularisation-based continual learning algorithms include (Ebrahimi et al., 2019; Park et al., 2019).

Another approach is the architecture-based one where the principal aim is to administer both the stability and adaptation issues via dividing the architecture into reusable parts that are less prone to changes, and other parts especially devoted to individual tasks (Rusu et al., 2016b; Fernando et al., 2017; Yoon et al., 2018; Du et al., 2019; He et al., 2019; Li et al., 2019a; Xu et al., 2019). To learn a new task in the work by Rusu et al. (2016a), the whole network from the previous task is first copied then augmented with a new part of the architecture. Although this is effective in eradicating catastrophic forgetting, there is a clear scalability issue since the architecture growth can be prohibitively high, especially with an increasing number of tasks. The work introduced in (Li et al., 2019b) bases its continual learning on neural architecture search, whereas the representation in (Javed & White, 2019) is optimised such that online updates minimize the error on all samples while limiting forgetting. The framework proposed by Xu & Zhu (2018) interestingly aims at solving

this neural architecture structure learning problem, while balancing the tradeoff between adaptation and stability, via designed reinforcement learning (RL) strategies. When facing a new task, the optimal number of neurons and filters to add to each layer is cast as a combinatorial optimisation problem solved by an RL strategy whose reward signal is a function of validation accuracy and network complexity. Another RL based framework is the one presented by Kaplanis et al. (2018) where catastrophic forgetting is mitigated at multiple time scales via RL agents with a synaptic model inspired by neuroscience. Bottom layers (those near the input) are generally shared among the different tasks, while layers near the output are task-specific. Since the model structure is usually divided a priori and no automatic architecture learning nor adaptation takes place, alteration on the shared layers can still cause performance loss on earlier tasks due to forgetting (Shin et al., 2017). A clipped version of maxout networks (Goodfellow et al., 2013) is developed in (Lin et al., 2018) where parameters are partially shared among examples. The method in (Ostapenko et al., 2019) is based a dynamic network expansion accomplished by a generative adversarial network.

The memory-based approach, which is the third influential approach to address the adaptation-catastrophic forgetting tradeoff, relies on episodic memory to store data (or pseudodata) from previous tasks (Ratcliff, 1990; Robins, 1993; 1995; Hattori, 2014; Rolnick et al., 2018; Teng & Dasgupta, 2019). A major limitation of the memory-based approach is that data from previous tasks may not be available in all real-world problems (Shin et al., 2017; Choi et al., 2019). Another limitation is the overhead resulting from the memory requirements, e.g. storage, replay, etc. In addition, the optimisation required to select the best observation to replay for future tasks is a source of further overhead (Titsias et al., 2019). In addition to the explicit replay form, some works have been based on generative replay (Thrun, 1996; Schmidhuber, 2013; Mocanu et al., 2016; Rebuffi et al., 2017; Kamra et al., 2017; Shin et al., 2017; van de Ven & Tolias, 2018; Wu et al., 2018). Notably, Shin et al. (2017) train a deep generative model based on generative adversarial networks (GANs, Goodfellow et al., 2014b; Goodfellow, 2016) to mimic past data. This mitigates the aforementioned problem, albeit at the added cost of the training of the generative model (Schwarz et al., 2018) and sharing its parameters. Alleviating catastrophic forgetting via replay mechanisms has also been adopted in reinforcement learning, e.g. (Isele & Cosgun, 2018; Rolnick et al., 2018). A similar approach was introduced by Lopez-Paz & Ranzato (2017) where gradients of the previous task (rather than data examples) are stored so that a trust region consisting of gradients of all previous tasks can be formed to reduce forgetting. Other algorithms based on replay mechanisms include (Aljundi et al., 2019a;b).

Equivalent tradeoffs to the one between adaptation and stability can be found in the literature since the work in (Carpenter & Grossberg, 1987), in which a balance was needed to resolve the stability-plasticity dilemma, where the latter refers to the ability to rapidly adapt to new tasks. The works introduced in (Chaudhry et al., 2018; Kim et al., 2019) shed light on the tradeoff between adaptation and stability, where they explore measures of intransigence and forgetting. The former refers to the inability to adapt to new tasks and data, whereas an increase in the latter clearly signifies an instability problem. Other recent works tackling the same tradeoff include (Riemer et al., 2019) where the transfer-interference (interference is catastrophic forgetting) tradeoff is optimised for the sake of maximising transfer and minimising interference by an algorithm based on experience replay and meta-learning. Other recent algorithms include the ORACLE algorithm by Yoon et al. (2019), which addresses the sensitivity of a continual learner to the order of tasks it encounters by establishing an order robust learner that represents the parameters of each task as a sum of task-shared and task-specific parameters. The algorithm in (Titsias et al., 2019) achieves functional regularisation by performing approximate inference over the function (instead of parameter) space. They use a Gaussian process obtained by assuming the weights of the last neural network layer to be Gaussian distributed. Our model is also related to the multi-task learning approach (Caruana, 1997; Heskes, 2000; Bakker & Heskes, 2003; Adel et al., 2017; Zhao et al., 2019a; Stickland & Murray, 2019; Zhao et al., 2019b).

## A.1 APPLICABILITY OF CLAW TO OTHER CL FRAMEWORKS

As mentioned in the main document, ideas of the proposed `CLAW` can be applied to continual learning frameworks other than VCL. The latter is more relevant for the inference part of `CLAW` since both are based on variational inference. As per the modeling ideas, e.g. the binary adaptation parameter depicting whether or not to adapt, and the maximum allowed adaptation, these can be integrated within other continual learning frameworks. For example, the algorithm in Xu & Zhu (2018) utilises reinforcement learning to adaptively expand the network. The optimal number of nodes and filters to

be added is cast as a combinatorial optimisation problem. In `CLAW`, we do not expand the network. As such, an extension of the work in (Xu & Zhu, 2018) can be inspired by `CLAW` where not only the number of nodes and filters to be added is decided for each task, but also a soft and more general version where an adaptation based on the same network size is performed such that the network expansion needed in (Xu & Zhu, 2018) can be further moderated.

## B   STATISTICAL SIGNIFICANCE AND STANDARD ERROR

In this section, we provide information about the statistical significance and standard error of `CLAW` and the competing continual learning frameworks. In Table 1, we list the average accuracy values (Figure 1 in the main document) obtained after completing the last two tasks from each of the six experiments. A bold entry in Table 1 denotes that the classification accuracy of an algorithm is significantly higher than its competitors. Significance results are identified using a paired t-test with p = 0.05. Each average accuracy value is followed by the corresponding standard error. Average classification accuracy resulting from `CLAW` is significantly higher than its competitors on the 6 experiments.

Table 1: Average test classification accuracy of the last two tasks in each of the six experiments: Permuted MNIST, Split MNIST, Split notMNIST, Split Fashion-MNIST, Omniglot and CIFAR-100, followed by the corresponding standard error. A bold entry denotes that the classification accuracy of an algorithm is significantly higher than its competitors. Significance results are identified using a paired t-test with p = 0.05. Average classification accuracy resulting from `CLAW` is significantly higher than its competitors on the 6 experiments.

| Classification Accuracy | CLAW | VCL | VCL + Coreset | EWC |
|---|---|---|---|---|
| Permuted MNIST (task 9) | **99.2** ± 0.2 % | 93.5 ± 0.3 % | 95.5 ± 0.3 % | 92.1 ± 0.4 % |
| Permuted MNIST (task 10) | **99.2** ± 0.1 % | 92.1 ± 0.3 % | 95 ± 0.5 % | 90.2 ± 0.4 % |
| Split MNIST (task 4) | **99.2** ± 0.2 % | 98.6 ± 0.3 % | 98.7 ± 0.2 % | 94.9 ± 0.4 % |
| Split MNIST (task 5) | **99.1** ± 0.2 % | 97.0 ± 0.4 % | 98.4 ± 0.3 % | 94.2 ± 0.5 % |
| Split notMNIST (task 4) | **98.7** ± 0.3 % | 95.8 ± 0.4 % | 96.9 ± 0.5 % | 92.9 ± 0.4 % |
| Split notMNIST (task 5) | **98.4** ± 0.2 % | 92.1 ± 0.3 % | 96.0 ± 0.3 % | 92.3 ± 0.4 % |
| Split Fashion-MNIST (task 4) | **93.2** ± 0.2 % | 90.0 ± 0.3 % | 90.7 ± 0.2 % | 89.4 ± 0.4 % |
| Split Fashion-MNIST (task 5) | **92.5** ± 0.2 % | 88.0 ± 0.2 % | 88.5 ± 0.4 % | 87.6 ± 0.3 % |
| Omniglot (task 49) | **84.5** ± 0.2 % | 81.1 ± 0.3 % | 81.8 ± 0.3 % | 78.2 ± 0.3 % |
| Omniglot (task 50) | **84.6** ± 0.3 % | 80.7 ± 0.3 % | 81.1 ± 0.4 % | 77.3 ± 0.3 % |
| CIFAR-100 (task 19) | **95.6** ± 0.3 % | 78.7 ± 0.4 % | 80.8 ± 0.3 % | 63.1 ± 0.5 % |
| CIFAR-100 (task 20) | **95.6** ± 0.3 % | 77.2 ± 0.4 % | 79.9 ± 0.4 % | 62.4 ± 0.4 % |
| Classification Accuracy | P&C | RCL | FRCL | LTG |
| Permuted MNIST (task 9) | 94.4 ± 0.3 % | 96.4 ± 0.5 % | 98.4 ± 0.4 % | 98.7 ± 0.3 % |
| Permuted MNIST (task 10) | 94.1 ± 0.6 % | 96.3 ± 0.3 % | 98.4 ± 0.5 % | 98.7 ± 0.3 % |
| Split MNIST (task 4) | 97.3 ± 0.5 % | 97.8 ± 0.7 % | 98.2 ± 0.3 % | 98.7 ± 0.2 % |
| Split MNIST (task 5) | 96.4 ± 0.4 % | 97.5 ± 0.6 % | 98.1 ± 0.2 % | 98.3 ± 0.3 % |
| Split notMNIST (task 4) | 97.8 ± 0.4 % | 97.7 ± 0.2 % | 96.1 ± 0.6 % | 97.8 ± 0.3 % |
| Split notMNIST (task 5) | 96.9 ± 0.5 % | 97.3 ± 0.5 % | 95.2 ± 0.7 % | 97.4 ± 0.3 % |
| Split Fashion-MNIST (task 4) | 91.4 ± 0.3 % | 91.1 ± 0.3 % | 90.4 ± 0.2 % | 92.5 ± 0.4 % |
| Split Fashion-MNIST (task 5) | 90.8 ± 0.2 % | 89.7 ± 0.4 % | 87.7 ± 0.4 % | 91.1 ± 0.3 % |
| Omniglot (task 49) | 82.8 ± 0.2 % | 80.1 ± 0.4 % | 79.9 ± 0.3 % | 83.6 ± 0.3 % |
| Omniglot (task 50) | 82.7 ± 0.3 % | 80.2 ± 0.4 % | 79.8 ± 0.5 % | 83.5 ± 0.3 % |
| CIFAR-100 (task 19) | 68.3 ± 0.6 % | 63.7 ± 0.6 % | 77.4 ± 0.7 % | 86.2 ± 0.4 % |
| CIFAR-100 (task 20) | 65.5 ± 0.6 % | 60.4 ± 0.6 % | 76.8 ± 0.6 % | 85.6 ± 0.5 % |

## C   OTHER EXPERIMENTAL DETAILS

Here are some additional details about the datasets in use:

The **MNIST** dataset is used in both the Permuted MNIST and Split MNIST experiments. The MNIST (Mixed National Institute of Standards and Technology) dataset (LeCun et al., 1998) is a handwritten digit dataset. Each MNIST image consists of $28 \times 28$ pixels, which is also the pixel size of the notMNIST and Fashion-MNIST datasets. The MNIST dataset contains a training set of 60,000 instances and a test set of 10,000 instances.

As mentioned in the main document, each experiment is repeated ten times. Data is randomly split into three partitions, training, validation and test. A portion of 60% of the data is reserved for training, 20% for validation and 20% for testing. Statistics reported are the averages of these ten repetitions.

Number of epochs required per task to reach a saturation level for CLAW (and the bulk of the methods in comparison) was 10 epochs for all experiments except for Omniglot and CIFAR-100 (15 epochs). Used values of $\omega_1$ and $\omega_2$ are 0.05 and 0.02, respectively.

For Omniglot, we used a network similar to the one used in (Schwarz et al., 2018), which consists of 4 blocks of $3 \times 3$ convolutions with 64 filters, followed by a ReLU and a $2 \times 2$ max-pooling. The same CNN is used for CIFAR-100. CLAW achieves clearly higher classification accuracy on both Omniglot and CIFAR-100 (Figures 1e and 1f).

## D    ABLATIONS

The plots displayed in this section empirically demonstrate how important the main adaptation parameters are in achieving the classification performance levels reached by CLAW. In each of the Figures 4a- 4f, the classification performance of CLAW is compared to the following three cases: 1) when the parameter controlling the maximum degree of adaptation is not learnt in a multi-task fashion, i.e. when the respective general value $s_{i,j}$ is used instead of $s_{i,j,t}$. 2) when adaptation always happens, i.e. the binary variable denoting the adaptation decision is always activated. 3) when adaptation never takes place. The differences in classification accuracy between CLAW and each of the other three plots in Figures 4a- 4f empirically demonstrate the relevance of each adaptation parameter.

## E    RUN-TIME

In Table 2, we report the wall-clock run time (in seconds) after finishing training in each of the six experiments: Permuted MNIST, Split MNIST, Split notMNIST, Split Fashion-MNIST, Omniglot and CIFAR-100. VCL and CLAW converge (i.e. reach the accuracy levels reported earlier) more quickly than the other methods. CLAW was the fastest in 3 out of the 6 experiments, whereas VCL was the fastest in the other 3, but their training run-time values have always been close to each other. As reported in the earlier sections, there is a significant difference in classification accuracy in favour of CLAW. This has been achieved within reasonably acceptable run-time levels thanks to the proposed design where the whole data-driven adaptation procedure is kept as part of an amortised variational inference algorithm. RCL is the slowest since it is based on reinforcement learning, where a large number of trials are typically required. This has also been acknowledged and reported therein (Xu & Zhu, 2018).

Table 2: Wall-clock run time (in seconds) after finishing training in each of the six experiments: Permuted MNIST, Split MNIST, Split notMNIST, Split Fashion-MNIST, Omniglot and CIFAR-100. As mentioned earlier, the statistics reported are averages of ten repetitions.

| **Training Time (in seconds)** | CLAW | VCL | VCL + Coreset | EWC | P&C | RCL | FRCL | LTG |
|---|---|---|---|---|---|---|---|---|
| Permuted MNIST (after 10 tasks) | 667 | 682 | 724 | 1117 | 1355 | 32575 | 919 | 705 |
| Split MNIST (after 5 tasks) | 649 | 637 | 708 | 1054 | 1312 | 31110 | 891 | 648 |
| Split notMNIST (after 5 tasks) | 722 | 714 | 792 | 1210 | 1407 | 34123 | 898 | 781 |
| Split Fashion-MNIST (after 5 tasks) | 829 | 818 | 901 | 1284 | 1498 | 35086 | 972 | 915 |
| Omniglot (after 50 tasks) | 1126 | 1241 | 1513 | 1637 | 1714 | 37247 | 1620 | 1312 |
| CIFAR-100 (after 20 tasks) | 792 | 810 | 914 | 802 | 1322 | 6102 | 1016 | 896 |

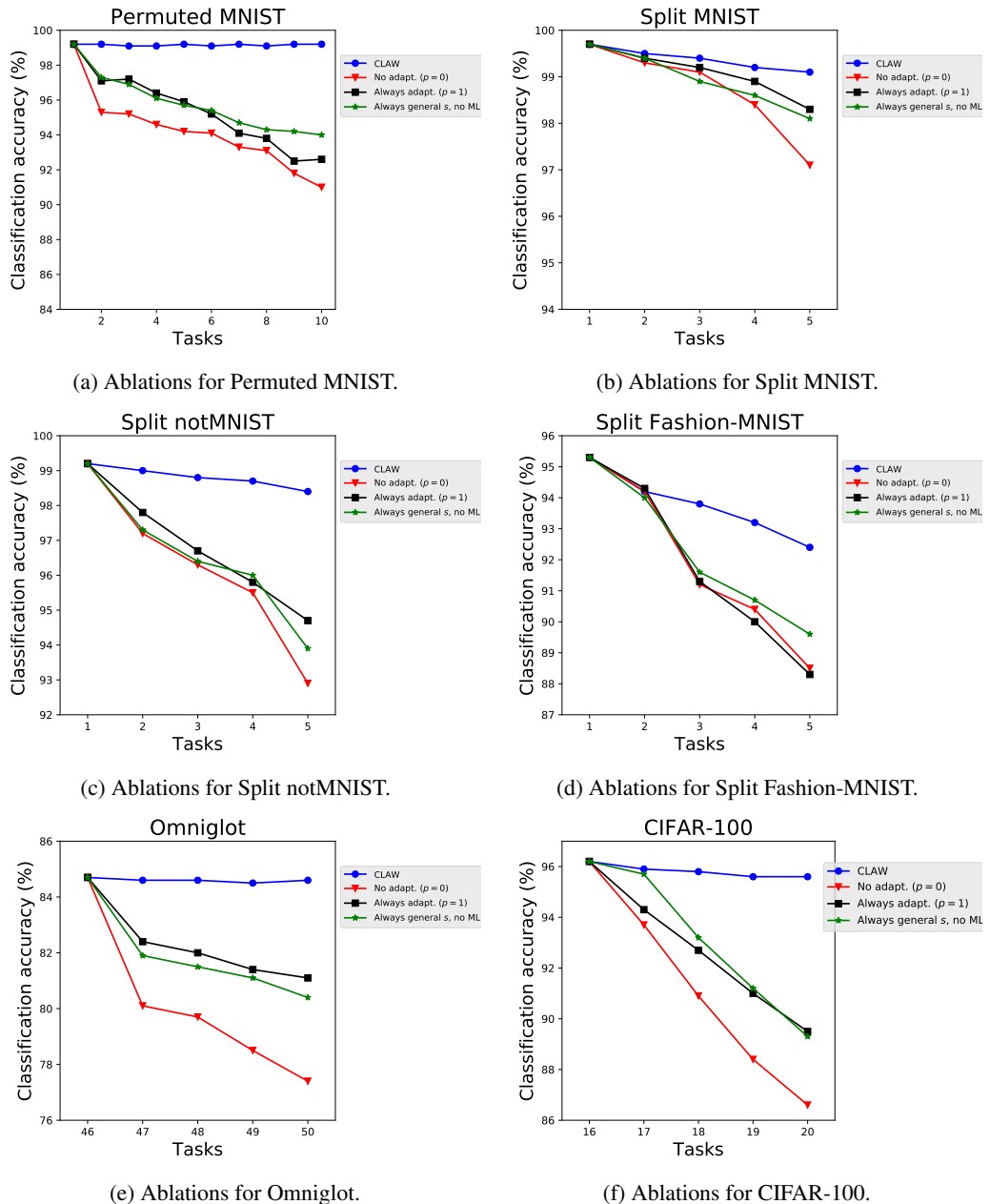

Figure 4: Ablation studies on different datasets.

