# OpenReview forum: "Continual Learning with Adaptive Weights (CLAW)"
_ICLR.cc/2020/Conference — Accept (Poster)_

### Official Review · AnonReviewer1 · 2019-10-23
**Official Blind Review #1**

**Rating:** 3

**Review:**

The paper proposed a novel probabilistic continual learning approach which automatically learn an optimal adaptation for arriving tasks while maintaining the performance of the past tasks. CLAW learns element-wise weight masking per task task with respect to the several learnable parameters.
CLAW fundamentally based on Variational Continual Learning, but it outperforms benchmarks on diverse dataset even without additional coreset. However, the ablation study and analysis on the model is weak and authors only show experimental observations. Also, the experiments are performed on old architectures. Then, it needs to show the model consistently outperform on recent deep network architectures, such as ResNet.


I have several questions,

- How about the training time / convergence rate of the CLAW compared to other methods?

- If the model need to divide the sample into two halves, isn't the model vulnerable when there are only a few number of samples with high variance? This situation is quite natural on realistic problem, like Imagenet.

- Why are the VCL variants with CNN not compared?

- There might be used a wrong plots in Figure 1 (e). It doesn't make sense that all methods show equal accuracy on task 46.



**Experience Assessment:**

I have published one or two papers in this area.

**Review Assessment: Checking Correctness Of Derivations And Theory:**

I assessed the sensibility of the derivations and theory.

**Review Assessment: Checking Correctness Of Experiments:**

I carefully checked the experiments.

**Review Assessment: Thoroughness In Paper Reading:**

I read the paper at least twice and used my best judgement in assessing the paper.

---

> ### Author Response · Authors · 2019-11-14
> **Response to R1**
>
> We thank the reviewer for the welcome feedback, which we will/do incorporate into the revised version.
>
> - ResNet:
> - A: We have tested the CLAW algorithm on standard architectural choices that are widely used in the continual learning literature in order to provide a fair comparison that does not conflate performance gains from improved feature extractors with performance gains from better continual learning.
>
> This is in line with many papers in continual learning (e.g. Progress & Compress, EWC, VCL, FRCL, etc) as well as similar paradigms like few-shot learning and meta-learning. This has been explicitly mentioned in previous seminal continual learning papers like the Progress & Compress algorithm by Schwarz et al., ICML 2018, and has also been explicitly discussed in few-shot learning papers like "Meta-learning probabilistic inference for prediction" by Gordon et al., ICLR 2019 and "A closer look at few-shot classification" by Chen et al., ICLR 2019. The latter discussed the role of deeper feature extractors in these learning scenarios. Of course, we also agree that combining CLAW with more powerful baseline architecture can further boost the performance.
>
>
> - Ablation study:
> - A: We have highlighted the introduction of an efficient inference mechanism which efficiently controls the following in a data-driven way: whether or not to adapt a neuron, and the magnitude of adaptation. In Section D in the Appendix, we have provided ablations exhibiting the relevance of each of the proposed adaptation parameters via comparing the performance of CLAW to the following cases:
>
> 1) When the magnitude of the maximum allowed adaptation is not learnt.
> 2) When adaptation always happens, i.e. the binary variable denoting the adaptation decision is always activated.
> 3) When adaptation never takes place.
>
> The results empirically demonstrate the relevance of each part and parameter of the proposed adaptation procedure. What else do you believe is missing?
>
>
>  - Training time:
> In Section C in the original version of the paper, we had provided the number of epochs of CLAW. In addition, we also provided the algorithmic complexity of CLAW near the end of Section 3 in the original version.
>
> We appreciate the benefits of looking at timings, though they are not often reported in related literature. We have just added to the revised version (Section E) values of the wall-clock run time (in seconds) after finishing training in each of the six experiments. VCL and CLAW converge more quickly than the other methods (more details can be found in Section E).
>
> We highlight the fact that one of the reasons why we were keen on designing the whole adaptation process to be part of the proposed, amortised variational inference procedure is the run-time efficiency.
>
>
>  - Dividing the sample into two halves:
> - A: One of the advantages of the VAE-based CLAW is the reduced variance (see e.g. Kingma and Welling, “Auto-encoding variational Bayes”, ICLR 2014 and Kingma et al., “Variational dropout and the local reparameterization trick”, NIPS 2015). This is supported by the small standard error values in Section B in the Appendix. This is empirically demonstrated via the results of a dataset like Omniglot. Omniglot consists of 50 realistic tasks depicting 50 alphabets, each consisting of handwritten characters. The small standard error values achieved by CLAW on Omniglot (as well as the other datasets) show that the model is not prone to this vulnerability.
>
>
> - Why are the VCL variants with CNN not compared?
> - A: This was because VCL does not originally enable the use of CNNs. As mentioned in Section 4, we have enabled the utilisation of CNNs in CLAW due to the adoption of the local reparameterisation trick. We have just adopted the local reparameterisation trick to VCL as well, and added these results to the revised version.
>
>
> - Figure 1 (e):
> - A: Thanks. We have fixed this typo in the revised version.

---

### Official Review · AnonReviewer2 · 2019-10-23
**Official Blind Review #2**

**Rating:** 8

**Review:**

This paper introduces CLAW, a complex but effective approach to continual learning with strong performance in the sequential task learning setting, as demonstrated on a number of standard benchmarks.

I recommend acceptance because:
- While conceptually similar to VCL, CLAW is convincingly shown to have superior performance across standard benchmarks and measures. The evaluation is thorough across the board, as far as I can tell.
- Forward transfer is shown to be substantially better compared to other methods. Experiments with long sequences of tasks (Omniglot, CIFAR-100) are particularly telling.
- Overall, the balance between not forgetting and still learning new tasks seems particularly favourable for the proposed method. This has been an elusive goal of continual learning research, hence the importance of accepting this work.


Here are some good reasons why an adversarial reviewer would reject this paper:
- The sequential task setting for continual learning has very little to add in practice, and to other branches of machine learning: it has little to say for domains where continual learning problems occur naturally, such as reinforcement learning, GAN training, multi-agent learning; all of these domains need continual learning solutions; while progress on standard benchmarks is important, we may be overfitting to these benchmarks.
- The paper is well written but the method is rather complex and presumably non-trivial to tune. This is actually characteristic of several top competing methods on these benchmarks; getting the last bit of performance seems to require this complexity, but it also makes it that much harder to generalize such methods beyond these benchmarks. For example, exploiting well partitioned datasets into different tasks and known task labels is a good starting point, but once such information is not available all bets are off. Acceptance means encouraging work on these benchmarks; is this really what we should do?
- It's perfectly tractable to store some small amount of old data for these problems in an 'episodic memory', so one could claim that an entire class of relevant baselines is missing, e.g. A-GEM, iCaRL, etc.


Luckily, I am not an adversarial reviewer, but I want to see progress across a more diverse and widely relevant set of continual learning challenges.

**Experience Assessment:**

I have published in this field for several years.

**Review Assessment: Checking Correctness Of Derivations And Theory:**

I did not assess the derivations or theory.

**Review Assessment: Checking Correctness Of Experiments:**

I assessed the sensibility of the experiments.

**Review Assessment: Thoroughness In Paper Reading:**

I read the paper at least twice and used my best judgement in assessing the paper.

---

> ### Author Response · Authors · 2019-11-14
> **Response to R2**
>
> We thank the reviewer for the welcome feedback, which we will/do incorporate into the revised version.
>
> - The sequential task setting for continual learning has very little to add in practice, and to other branches of machine learning: it has little to say for domains where continual learning problems occur naturally, such as reinforcement learning, GAN training, multi-agent learning; all of these domains need continual learning solutions:
>
> - A: As mentioned in the paper, our architecture as well as the modeling ideas can be utilised in the multi-task learning paradigm.
>
> In addition, with fairly simple changes, the methods presented in the paper can be generalised from the classification setting to some of the paradigms mentioned by the reviewer, namely conditional generation in GANs, reinforcement learning (RL) and generally environments that change over time (e.g. due to the behaviour of other agents, etc).
>
> We also intend to extend the ideas presented in this paper in some of these learning paradigms in future work. One of the key differences between continual learning and some of these paradigms is that in the former the learner is aware of the boundaries between tasks, unlike with RL for instance. We have already begun to work on extensions of CLAW based on the harder case where the learner needs to discover such boundaries based on the uncertainty. In other words, and similar to what was developed by Farquhar and Gal, "Towards robust evaluations of continual learning", 2018, we test the ability of model uncertainty to distinguish a task boundary. Thanks to our variational inference modeling framework, there is potential to use uncertainty estimates to discover task boundaries, which is one step in the direction of sequential decision-making paradigms other than continual learning.
>
>
> - Complexity of CL algorithms:
> - A: This is another very good point. This is one of the reasons we have been keen on developing and presenting the ideas in a “modular” framework, separating the modeling from the inference ideas. This eventually makes it straightforward to improve one particular module in the future, due to its minimal entanglement with the other modules of the framework. We also believe that one of the advantages of shedding light on the relevance of each of the principal parameters (like what we empirically provided via the ablative analysis in Section D in the Appendix) is to make the overall presentation more unequivocal.
>
>
> - 'episodic memory' baselines:
> - A: Methods developed here are orthogonal to memory-based methods, and it would therefore be simple to combine with them. In principle, this is similar to the coreset version of VCL (the coreset is analogous to an episodic memory that retains important training data points from previous tasks). The inference framework in CLAW also naturally allows for the integration and storage of an episodic memory. We have added this clarification to the text.
>
>
> - Progress across a more diverse and widely relevant set of continual learning challenges:
>  - A: We agree that in general the current baselines in continual learning could be improved and made more real-world relevant. We have tried to provide a thorough analysis: we have performed a range of experiments in line with previously published relevant papers, and have aimed at carrying out a set of experiments that is richer than the average (compared to our 5 examined datasets, some of the seminal, top-tier publications in the CL literature considered fewer datasets, for example EWC: 2 datasets, VCL: 3 datasets, RCL: 3 datasets & LTG: 3 datasets).
>
> We agree with your point. As mentioned earlier, we hope to make further use of the uncertainty estimates and their high fidelity to move on to other sequential decision-making paradigms with no task boundaries. The more stringent assumption that the learner is not aware of when the changes occur in the task distribution has also been highlighted by the seminal continual learning paper “Progress and compress”, ICML 2018, as a cornerstone in the direction of moving from continual learning to more real-life problems (and also a future objective of theirs), which we believe that we have a strong basis of thanks to the variational inference foundation of CLAW.

---

### Official Review · AnonReviewer3 · 2019-10-25
**Official Blind Review #3**

**Rating:** 3

**Review:**

The authors propose a new continual learning method. The model is based on the probabilistic model and variational inference. To show the effectiveness of the proposed model, the authors do experiments on several benchmarks.

The model lacks enough technical novelty. The motivation of this paper is a task-specific weight adaptation mechanism, which seems a simple version of [1]. In addition, the design of b^T as s/(1+e^{-a^T})-1 is not well-motivated. It is better to explain more.

In addition, if the authors compare the performance on both negative transfer (forgetting) and forward transfer (performance on the new task). I suggest the authors compare with [2], which also focuses on the performance of new tasks.

Minor:
1. The font of figures and its legends are small.

[1] Li, Xilai, et al. "Learn to Grow: A Continual Structure Learning Framework for Overcoming Catastrophic Forgetting." ICML’19.

[2] C Finn et al. “Online Meta-learning” ICML’19


**Experience Assessment:**

I have published one or two papers in this area.

**Review Assessment: Checking Correctness Of Derivations And Theory:**

I assessed the sensibility of the derivations and theory.

**Review Assessment: Checking Correctness Of Experiments:**

I assessed the sensibility of the experiments.

**Review Assessment: Thoroughness In Paper Reading:**

I read the paper at least twice and used my best judgement in assessing the paper.

---

> ### Author Response · Authors · 2019-11-14
> **Response to R3**
>
> We thank the reviewer for the welcome feedback, which we will/do incorporate into the revised version.
>
> - The motivation of this paper is a task-specific weight adaptation mechanism, which seems a simple version of [1]:
> - A: We respectfully disagree. The methodology we propose in CLAW is completely different from [1], i.e. "Learn to Grow". They both try and adapt the architecture, and use similar terminology for the "reuse", "adaptation" and "new" options. However, the methodologies used to achieve the adaptation in the two papers are completely different. CLAW introduces a novel algorithm where the adaptation itself becomes part of the variational inference procedure, whereas “Learn to Grow” is based on neural architecture search. CLAW is the first method to offer CL architecture adaptation without having to expand the architecture with new layers or new neurons. In the beginning (regardless of the number of tasks), our framework adds solely (a total of) 3 parameters per neuron, and from this point onwards there is no need to add any further parameters along with the introduction of new CL tasks (no matter how many tasks).
>
> - Motivation of the design of b^T as s/(1+e^{-a^T})-1:
> - A: The variable b^T constrains the range of the neuron adaptation. This is useful to limit overfitting, especially when dealing with noisy data. Instead of keeping the adaptation values unconstrained, more sensitive to noise and prone to overfitting, the unconstrained adaptation variable a^T is constrained with the corresponding b^T. As such, rather than varying in the interval [-inf, inf], the range of adaptation values becomes [0, s], which also facilitates the optimisation. Though in a different context, this design is similar to Swietojanski and Renals [3] (reference added to the revised version). As was already mentioned in our original version, the addition of $1$ is to facilitate the usage of a probability distribution while still keeping an adaptation range allowing for the attenuation or amplification of each neuron's contribution.
>
> The (differentiable) logistic function is well-known to be powerful and flexible in bounding (constraining) its domain within a certain range while still counting on a relatively low number of adaptation parameters. We empirically tried other options including: rigidly projecting the unconstrained adaptation values into a certain interval (e.g. [-s, s]), using a logistic function without learning its maximum s (as in the ablative analysis in Section D), and using a fixed maximum (e.g. 2/(1+e^{-a^T})-1). Of these options, and as expected, using the logistic function empirically proved to be more effective.
>
>  [3] P. Swietojanski and S. Renals. Learning hidden unit contributions for unsupervised speaker adaptation of neural network acoustic models. IEEE Spoken Language Technology Workshop (SLT), 2014.
>
> - Comparison with [2]:
> - A: We have performed comparisons with the interesting "online meta-learning" algorithm (to be added to the revised version). CLAW performs significantly better in terms of classification accuracy, achieved reduction in catastrophic forgetting and the achieved degree of positive forward transfer. Having said that, we also have to highlight the fact that the principal goals of the “online meta-learning” paper are rather different since they are focussed around meta-learning and ensuring that online meta-learning, and especially the follow the meta-leader (FTML) algorithm, can be successfully applied to non-stationary learning problems.
> Here are the results of the comparisons between CLAW and the FTML algorithm presented in the online meta-learning paper:
>
> Classification accuracy:
>                                              | CLAW       FTML
> CIFAR-100 (after 19 tasks) | 95.6%      81.3%
> CIFAR-100 (after 20 tasks) | 95.6%      80.2%
>
> Omniglot (after 49 tasks)  | 84.5%      79.7%
> Omniglot (after 50 tasks)  | 84.6%      78.5%
>
> Catastrophic forgetting:
>                                              | CLAW      FTML
> CIFAR-100 (after 19 tasks)  | 95.4%     82.4%
> CIFAR-100 (after 20 tasks)  | 95.4%     81.8%
>
> Omniglot (after 49 tasks)   | 75.2%     68.4%
> Omniglot (after 50 tasks)   | 75.1%     68%
>
> Positive forward transfer:
>                                                       | CLAW      FTML
> CIFAR-100 (after 18 other tasks)| 95.3%     78.6%
> CIFAR-100 (after 19 other tasks)| 95.6%     78.7%
>
> Omniglot (after 48 other tasks)   |   85.2%      79.4%
> Omniglot (after 49 other tasks)   |   86.1%      79.7%
>
> We acknowledge that this is one of the methods that one can compare to, and thank the reviewer for pointing it out. However, in the original version of the paper, we have compared to 7 CL algorithms. We believe this represents a thorough analysis that goes beyond the comparisons found in many published papers (for instance EWC, VCL and P&C compared theirs with 2, 4 and 4 other algorithms, respectively), although we acknowledge there is a rapid growth of literature in this area.

---

### Public Comment · ~Prakhar_Kaushik1 · 2020-02-04
**Results on Permuted MNIST**

I don't know the exact process the authors followed for training permuted mnist task. If they used Zenke et. al like architecture (2 100x100 hidden layers) and 20 epochs or less of training - an individual task by itself barely reaches 97-98% test accuracy under the mentioned optimization parameters.
The code isn't available in the public domain and seeing results of this Deepmind paper on the same experiments- https://arxiv.org/pdf/1901.11356.pdf makes me wonder how the authors actually went about the experimentation. For e.g. the same method's results (FRCL) are overstated in your results than the original work. The authors should clarify what additional tricks, if any, they used for such 'boost' in performances of all methods.
Surprisingly, none of the reviewers seem to notice it either.

---

> ### Author Response · Authors · 2020-02-05
> **Resp.**
>
> The same architecture (as Zenke et al.) was used as a basis to enable fair comparison. Apparently, the method's characteristics lead to changes in how the parameters are computed and therefore the results change. Results of 98% in Zenke et al. improved to 99% in CLAW, based on the same architecture. This is similar to the improvement by VCL with coresets on the same data; the results therein were higher than what was originally reported in Zenke et al. . CLAW goes one step further.
> Regarding the performance on the first task on Permuted MNIST, this dataset follows a random permutation process. Zenke et al. used the same architecture but the outcome of the random permutation process is not necessarily the same. What matters is that the same outcome of the permutation process is used here throughout all the compared methods.

---

### Decision · Program_Chairs · 2019-12-19

**Decision:**

Accept (Poster)

**Comment:**

The paper proposes a new variational-inference-based continual learning algorithm with strong performance.

There was some disagreement in the reviews, with perhaps the one shared concern being the complexity of the proposed method. One reviewer brought up other potentially related work, but this was convincingly rebutted by the authors. Finally, one reviewer had an issue with the simplicity with the networks in the experiments, but the authors rightly pointed out that the architectures were simply designed to match those from the baselines.

Continual learning has been an active area for quite some time and convincingly achieving SOTA in a new way is a strong contribution, and will be of interest to the community. Progress in a field is sometimes made by iteratively simplifying an initially complex solution, and this work lays in a brick in that direction. For these reasons, I recommend acceptance.